# Glucose Limitation Sensitizes Cancer Cells to Selenite-Induced Cytotoxicity via SLC7A11-Mediated Redox Collapse

**DOI:** 10.3390/cancers14020345

**Published:** 2022-01-11

**Authors:** Hui Chen, Han Zhang, Lixing Cao, Jinling Cui, Xuan Ma, Chong Zhao, Shutao Yin, Hongbo Hu

**Affiliations:** Beijing Key Laboratory for Food Non-Thermal Processing, College of Food Science and Nutritional Engineering, China Agricultural University, Beijing 100083, China; B20183060527@cau.edu.cn (H.C.); S20193060988@cau.edu.cn (H.Z.); dolly1510505650@126.com (L.C.); cuijinling0420@cau.edu.cn (J.C.); BS20193060576@cau.edu.cn (X.M.); zhaoch0206@cau.edu.cn (C.Z.); yinshutao@cau.edu.cn (S.Y.)

**Keywords:** selenite, glucose limitation, SLC7A11, NADPH, ROS

## Abstract

**Simple Summary:**

Selenite, a representative inorganic form of selenium, is preferentially accumulated in tumors. The therapeutic potential of sodium selenite in tumors has received significant attention. However, the effect of sodium selenite in the treatment of established tumors is hampered by its systemic toxicities. In this study, we found selenite exerted a stronger lethality to the cancer cells under the condition of glucose limitation in vitro and an enhanced inhibitory effect on tumor growth when combined with intermittent fasting in vivo. In addition, this treatment showed no obvious toxicity to normal cells and mice. The findings of the present study provide an effective and practical approach for increasing the therapeutic window of selenite and imply that combination of selenite and fasting holds promising potential to be developed a clinically useful regimen for treating certain types of cancer.

**Abstract:**

Combination of intermittent fasting and chemotherapy has been drawn an increasing attention because of the encouraging efficacy. In this study, we evaluated the anti-cancer effect of combination of glucose limitation and selenite (Se), a representative inorganic form of selenium, that is preferentially accumulated in tumors. Results showed that cytotoxic effect of selenite on cancer cells, but not on normal cells, was significantly enhanced in response to the combination of selenite and glucose limitation. Furthermore, in vivo therapeutic efficacy of combining selenite with fasting was dramatically improved in xenograft models of lung and colon cancer. Mechanistically, we found that SLC7A11 expression in cancer cells was up-regulated by selenite both in vitro and in vivo. The elevated SLC7A11 led to cystine accumulation, NADPH depletion and the conversion of cystine to cysteine inhibition, which in turn boosted selenite-mediated reactive oxygen species (ROS), followed by enhancement of selenite-mediated cytotoxic effect. The findings of the present study provide an effective and practical approach for increasing the therapeutic window of selenite and imply that combination of selenite and fasting holds promising potential to be developed a clinically useful regimen for treating certain types of cancer.

## 1. Introduction

The combination of intermittent fasting (IF)/fasting-mimicking diet (FMD) and chemotherapy has been drawn an increasing attention. Accumulating evidence indicates that this combination treatment can not only increase cancer therapeutic effect, but also reduce detrimental effects of chemotherapy on normal cells [1,2,3,4]. Therefore, IF/FMD is proposed as a promising strategy to improve therapeutic efficacy and prevent side effects. This differential effect on cancer and normal cells is attributed to the different adaptability of cancer and normal cells to the condition of starvation [5]. Cancer cells are more vulnerable to the starvation condition partially due to their aberrant metabolic attributes. For example, the Warburg effect is a key feature of cancer cell metabolism. Shortage in the availability of glucose by fasting forces a shift from aerobic glycolysis (Warburg effect) to mitochondrial oxidative phosphorylation in cancer cells to meet the requirement for cancer cell growth [1,6]. The increased oxidative phosphorylation results in elevated ROS generation, which in turn renders cancer cells susceptible to the chemotherapy [7,8]. A recent study by Liu et al. [9] revealed the involvement of SLC7A11 in vulnerability of cancer cells to glucose starvation. SLC7A11, a cystine/glutamate antiporter, is responsible for uptake of extracellular cystine to maintain cellular redox balance. SLC7A11 is overexpressed in many types of cancers including lung cancer and triple-negative breast cancer [10,11]. Overexpression of SLC7A11 promotes tumor development via suppressing oxidative stress-induced ferroptosis and non-ferroptotic cell death [12,13]. On the other hand, overexpression of SLC7A11 leads to increase of cystine uptake, accompanied by augmentation of glutamate export and NADPH consumption (due to NADPH-dependent reduction of cystine to cysteine), which forces cancer cells to be highly dependent on pentose phosphate pathway (PPP) [9]. Such metabolic feature confers the cancer cells with elevated expression of SLC7A11 more sensitive to glucose deprivation owing to cystine accumulation-mediated NADPH depletion and impairment of cellular redox homeostasis.

Selenium is an essential micronutrient with multiple biological functions. Among them, the anticancer activity has been extensively investigated and yield controversial outcomes. It is generally believed that the anticancer activity of selenium is associated with dose levels, forms and nutrient status of the body [14]. Selenite, a representative inorganic form of selenium, is preferentially accumulated in tumors [15,16]. Preclinical and clinical studies have shown that selenite alone or in combination with chemotherapy or radiotherapy is effective against a variety of cancer types [17,18,19,20,21]. Mechanistically, induction of reactive oxygen species and activation of p53 play critical role in its cytotoxic and sensitization effect [22]. Recent studies revealed that selenium enrichment in tumor is closely associated with the elevated expression of SLC7A11, supporting an important role of SLC7A11 in the selenophilic properties of cancer cells [10,11].

As discussed above, elevated expression of SLC7A11 renders cancer cells more susceptible to glucose limitation due to redox imbalance. We hypothesized that SLC7A11 might be up-regulated in response to selenite challenge and, therefore, combining selenite with glucose starvation could achieve synergistic induction of oxidative stress and cytotoxic effect on cancer cells. This hypothesis was tested in the present study by using both in vitro and in vivo models.

## 2. Materials and Methods

### 2.1. Chemicals and Reagents

Sodium selenite (Se), methylseleninic acid (MSeA), dihydroethidium (DHE), 2′,7′-dichlorodihydrofluorescein diacetate (DCFH-DA), glutamine (Gln), glucose and manganese (III) tetrakis (1-methyl-4-pyridyl) porphyrin (MnTMPyP) were purchased from Sigma Chemical Co. (St. Louis, MO, USA). BAY-876, deferoxamine mesylate (DFO) and ferrostatin-1 (Fer-1) were purchased from MedChem Express (Monmouth Junction, NJ, USA). Monosodium glutamate (MSG), salicylazosulfapyridine (SAS) and 2-deoxy-D-glcose (2-DG) were purchased from Aladdin (Shanghai, China). FerroOrange and Liperfluo were purchased from DOJINDO (Kyushu, Japan). Antibodies specific for SLC7A11 (12691), SLC7A11 (98051) and β-actin (3700) were purchased from Cell Signaling Technology (Beverly, MA, USA). The secondary antibodies: Horseradish peroxidase-linked Goat Anti-Rabbit IgG and Horseradish peroxidase-linked Goat Anti-Mouse IgG were obtained from MBL International Corporation (Beijing, China). SLC7A11 small interfering RNA (siRNA) and nontargeting siRNA were purchased from Santa Cruz Biotechnology (Santa Cruz, CA, USA).

### 2.2. Cell Culture and Treatments

HCT116 human colon cancer cells, LLC lung cancer cells, HepG2 liver cancer cells, MCF-7 and MDA-MB-231 breast cancer cells, HK2 normal kidney cells and MRC-5 normal lung cells were grown in Dulbecco’ modified Eagle’s medium (DMEM) supplemented with 10% fetal bovine serum (FBS), 2 mM L-glutamine and 25 mM glucose unless otherwise indicated. A549 lung cancer cells, DU145 prostate cancer cells were grown in RPMI medium supplemented with 10% FBS, 2 mM L-glutamine and 10 mM glucose unless otherwise indicated. DMEM medium and RPMI-1640 medium were purchased from Procell Life Science &Technology Co., Ltd. (Wuhan, China), these mediums contained no glucose and glutamine. Therefore, the glucose and glutamine concentrations could be adjusted to indicated concentration in our experiments. FBS was purchased from Biological Industries (Israel). For glucose limitation experiments, cells were washed three times with PBS pH 7.2 and then incubated in the indicated glucose conditions. All cultures were maintained in a humidified tissue culture incubator at 37 °C in 5% CO_2_. It should be noted that glutamine concentration was 2 mM under the condition of only glucose limitation experiments and glutamine deprivation experiments were proceeded in the DMEM medium with low glucose concentration of 2.5 mM.

### 2.3. Crystal Violet Staining

Cells (1 × 10^5^ cells/well) were grown on sterile coverslips embedded in a 12-well plate. The next day, the medium was changed to indicated glucose-containing DMEM for different treatments. After treatment for indicated times, the culture medium was aspirated and replaced with 1% glutaraldehyde solution for 15 min. Then, the cells were stained with a 0.02% crystal violet solution for 30 min. After that, the solution was replaced by 70% ethanol for solubilization. The OD value at 570 nm was measured by microplate reader.

### 2.4. Cell Death Evaluation

Cells (2 × 10^5^ cells/well) were seeded on sterile coverslips embedded in a 6-well plate. The next day, the medium was changed to indicated glucose-containing DMEM for different treatments. After 24 h, the cells were collected and resuspended in PBS with Annexin V-FITC and PI and then incubated at room temperature for 15 min and analyzed by Becton Dickinson FACSCalibur Flow Cytometer at an excitation wavelength of 488 nm. Ten thousand cells were collected from the analyzed sample. The cells of Annexin V positive and PI negative represented early apoptotic cells, the cells of both Annexin V and PI positive represented late apoptotic cells and the cells positive for PI only represented necrotic cells. The percentage of cell deaths was calculated by adding up early apoptotic cells, late apoptotic cells and necrotic cells and dividing the total cell number.

### 2.5. Determination of Cell ROS

Cells (2 × 10^5^ cells/well) were seeded on sterile coverslips embedded in a 6-well plate. The next day, the medium was changed to indicated treatments. Reactive oxygen species (ROS) measurement using DHE and DCFH-DA staining was performed as described as follows: cells were trypsinized, washed with PBS and then resuspended in the medium without FBS and loaded with 10 μM DHE or 20 μM DCFH-DA and then incubated for 30 min at 37 °C in the dark. Afterward, cells were washed with PBS and resuspended in the PBS used for fluorescence analysis by flow cytometer. Fluorescence increase was estimated utilizing the wavelengths 535 nm (excitation) and 610 nm (emission) for DHE and wavelengths 485 nm (excitation) and 535 nm (emission) for DCFH-DA.

### 2.6. GSH, NADP+, NADPH and Cysteine Measurement

Cells (2 × 10^5^ cells/well) were seeded on sterile coverslips embedded in a 6-well plate. The next day, the medium was changed to indicated treatments. Glutathione (GSH) in cell was measured using a commercial kit from Nanjing Jiancheng Bioengineering Institute (Nanjing, China) following the manufacturer’s instruction. The intracellular levels of NADPH and total NADP (NADPH+NADP+) were measured according to the protocol of manufacturer from DOJINDO (Kyushu, Japan). The quantification of cysteine was determined using Cysteine assay kit (Abcam, Cambridge, UK).

### 2.7. Cystine Measurement by LC-MS

Cells (1.5 × 10^6^ cells/well) were seeded on 75 cm culture flasks. The next day, the medium was changed to indicated treatments. Cells were trypsinized, washed twice with ice cold PBS, extracted by adding 0.3 mL methanol-water (*v*/*v* = 1:1) mixture at ice temperature and then disrupted by ultrasound. Cell debris was pelleted by centrifugation at 12,000 rpm for 10 min at 4 °C and the supernatant was transferred to a fresh tube. A total of 20 μL cell extracts, 5 μL internal standard substance (canocinal amino acid mix with 1.23 mM L-Cystine-13C2-15N1, Cambridge Isotope Laboratories) and 40 μL isopropyl alcohol-formic acid (*v*/*v* = 99:1) were mixed and vortex oscillated for 2 min, then centrifugated at 12,000 rpm for 10 min at 4 °C. 10 μL supernatant were derivatized according to the protocol of AccQ Tag kit (Waters, Milford, MA, USA) and then analyzed by LC-MS. Analyte concentrations were quantified by comparison to standard curves of cystine prepared by the same method. To determine intracellular concentrations, the concentrations of cell protein in the same volume of cell resuspension were determined.

LC-MS analysis was performed as follows: the LC-MS system consisted of a Waters UPLC I-Class system and Waters XEVO TQ-XS quadrupole rods tandem mass spectrometer equipped with a ESI probe. Mass data acquisition and remote control of the LC-MS system were done by Masslynx software. Chromatography was performed with a waters UPLC HSS T3(150 × 2.1 mm, 1.8 μm particle size). The mobile phase consisted of solvent A (0.1% formic, water) and solvent B (acetonitrile and water, *v*/*v* = 95:5) with a gradient elution (0–0.5 min, 96–96% A; 0.5–2.5 min, 96–90% A; 2.5–5 min, 90–72% A; 5–6 min, 72–5% A; 6–7 min, 5–5% A; 7–7.1 min, 5–96% A; 7.1–9 min, 96–96% A). The flow rate of the mobile phase was 0.5 mL/min. The column temperature was maintained at 50 °C. The injection volume was 5 μL. The exactive was operated in positive ionization mode with an electrospray ionizatio interface. The instrument parameters were as follows: positive ESI source temperature, 50 °C; capillary voltage, 1.5 kV; cone voltage, 20 V; cone gas flow, 150 L/Hr; desolvation gas flow, 1000 L/Hr.

### 2.8. LPO, LIP Measurement

Cells (2 × 10^5^ cells/well) were seeded on sterile coverslips embedded in a 6-well plate. The next day, the medium was changed to indicated treatments. The total cellular labile iron pool (LIP) and lipid peroxidation (LPO) measurement using FerroOrange and Liperfluo were performed as described as follows: cells were trypsinized, washed with PBS and then resuspended in the PBS and loaded with 1 μM FerroOrange and or 1 μM Liperfluo and then incubated for 30 min at 37 °C in the dark. Then, the cells were analyzed for fluorescence by flow cytometer. Fluorescence increase was estimated utilizing the wavelengths 561 nm (excitation) and 593 nm (emission) for FerroOrange and the wavelengths 488 nm (excitation) and 535 nm (emission) for Liperfluo.

### 2.9. RNA Interference

Cells were transfected with 40 nM SLC7A11 siRNA or nontargeting siRNA using the INTERFERin siRNA transfection reagent according to the manufacturer’s instructions (Polyplus-Transfection, Inc., New York, NY, USA) and then were used for subsequent experiments.

### 2.10. Western Blotting

Western blotting was performed according to the method of Yan et al. [23] with minor modifications. The cell lysate was prepared in ice-cold radioimmunoprecipitation assay (RIPA) buffer. Cell lysates were resolved by electrophoresis and transferred to a polyvinylidene fluoride (PVDF) membrane (Millipore, Billerica, MA, USA; IPVH00010). The blot was then probed with primary antibody followed by incubation with the appropriate horseradish peroxidase-conjugated secondary antibodies. The signal was visualized by enhanced chemiluminescence (Fisher/Pierce, Rockford, IL, USA; 32106) and recorded on an X-ray film (Eastman Kodak Company, Rochester, NY, USA; XBT-1). The whole western blot figures can be found in Appendix A.

### 2.11. Xenograft Tumor Models

The animals were housed under specific pathogen-free conditions at 22 ± 2 °C with 55 ± 10% relative humidity and with 12 h day/light cycles. All experiments were performed in accordance with the guidelines established in the Principles of China Agricultural University Institutional Animal Care and Use Committee. For xenograft experiments, 6–8-week-old male C57 BL/6N mice from Charles River (Beijing, China) were subcutaneously injected with 2 × 10^6^ LLC cells resuspended in 100 μL of PBS; 5-week-old male BALB/c nude mice from Charles River (Beijing, China) were subcutaneously injected into the dorsal side with 3 × 10^6^ HCT116 cells resuspended in 100 μL of PBS. When tumors were palpable (5 days after inoculation), mice were randomly divided into different experimental groups. Mice were kept on the feeding/fasting protocols performed as described as the reported [1]. In short, fasting cycles were achieved by complete removal of food while allowing free access to water for 24 h from 6 pm to 6 pm of the following day when food was re-supplied ad libitum. Selenite dissolved in water at the dose of 2 mg/kg body weight was administered every 48 h at 9 am (time in fasting cycle) via oral gavage. Body weights were recorded every 2 days and tumor volumes were measured every 2 days by a digital vernier caliper according to the following equation: tumor volume (mm^3^) = (length × width^2^) × 0.5, where the length and width were expressed in millimeters.

### 2.12. Statistical Analysis

Data were presented as the mean ± SD. These data were analyzed by ANOVA with appropriate post hoc comparisons among means. *p* < 0.05 was considered statistically significant.

## 3. Results

### 3.1. Glucose Limitation Sensitizes Cancer Cells to Selenite-Mediated Cytotoxic Effect

To investigate influences of glucose on the anticancer activity of selenite, HCT116 human colon cancer cells were cultured in the medium with different levels of glucose and exposed to various concentrations of selenite (0, 1, 2.5 and 5 μM) for 24 h and the changes of cell viability were measured by crystal violet staining. As shown in Figure 1A, selenite-mediated cytotoxic effect was dramatically enhanced when the concentration of glucose decreased to 2.5 mM. This enhancement was further validated by measurement of cell death induction using Annexin V/PI staining and results were shown in Figure 1B. Consistent with the cell viability changes, cell death induction by selenite in HCT116 cells was significantly increased in the context of glucose deprivation (refers to 2.5 mM). Glucose is primarily taken up by cancer cells via GLUT1, a member of glucose transporter family and commonly overexpressed in cancer cells. Inhibition of GLUT1 by its inhibitor is supposed to cause intracellular glucose reduction. We next examined effect of GLUT1 inhibitor BAY-876 on selenite-induced cytotoxicity of HCT116 cells and results demonstrated that an enhanced cytotoxicity was achieved by all the combinations of selenite and BAY-876, further supporting the sensitization effect of glucose deprivation on selenite-induced cytotoxic effect against HCT116 colon cancer cells (Figure 1C). To determine whether the sensitization effect of glucose limitation on selenite-induced cytotoxicity was specific for HCT116 colon cancer cells, LLC and A549 lung cancer cells, HepG2 liver cancer cells, MCF-7 and MDA-MB-231 breast cancer cells and DU145 prostate cancer cells were tested. As shown in Figure 1D, the sensitization effect was also found in all additional cancer cell lines tested, indicating general applicability of the sensitization effect. Furthermore, we found that cell lines tested in the present study showed different sensitivity to glucose limitation. MDA-MB-231, HepG2, MCF7 and A549 cells were relatively insensitive to glucose limitation in comparison with LLC cells. For these cell lines, a significant reduction of cell viability was detected when the cells were challenged with glucose limitation for an additional 12 h (data not shown).

To determine if the sensitization effect was restricted to cancer cells, MRC-5 normal lung cells and HK-2 normal kidney cells were tested and results showed that the sensitization effect was not observed in these two normal cell lines, suggesting that glucose starvation specifically potentiated cancer cells but not normal cells to selenite-mediated cytotoxicity. The anticancer activity of selenium is closely associated with its forms. We asked if the sensitization effect of glucose limitation on selenite can be also achieved with other forms of selenium compounds. Influence of glucose limitation on the cytotoxicity of methylseleninic acid (MSeA), a representative of organic selenium compounds, was assessed and results showed that glucose starvation failed to potentiate HCT116 colon cancer cells to MSeA-mediated cytotoxic effect (Figure 1E), suggesting that the sensitization effect of glucose deprivation on selenium compounds was form-dependent.

### 3.2. The Sensitization Effect of Glucose Limitation on Selenite Is Attributed to Elevated ROS Generation

Generation of ROS, mainly superoxide, plays a critical role in selenite-mediated cytotoxic effect on cancer cells [24,25,26]. We hypothesized that the enhanced cytotoxic effect by combination of selenite and glucose deprivation might be associated with boosted ROS generation. The changes of selenite-induced ROS in response to different concentrations of glucose were measured by flow cytometry following staining with DHE or DCFH-DA in HCT116 colon cancer cells. As shown in Figure 2A, at concentration of 2.5 mM glucose, selenite-induced ROS was significantly higher than that found at concentrations of 10 mM and 25 mM glucose, indicating glucose deprivation promoted ROS generation in response to selenite. To assess the role of elevated ROS in the enhanced cytotoxicity by combining selenite with glucose deprivation, we evaluated effect of ROS scavenger MnTMPyP on the enhanced cytotoxicity in HCT116 cells. As shown in Figure 2B, the cytotoxicity induced by combination of selenite and glucose deprivation was nearly abolished in the presence of MnTMPyP. In agreement with the cytotoxicity inhibition, ROS induction by either selenite alone or in combination with glucose deprivation was completely scavenged by the antioxidant (Figure 2C). In addition, similar results were also found in LLC lung cancer cells (Figure 2D,E). The results clearly suggested that the boosting ROS generation by combination of selenite and glucose deprivation contributed to the enhanced cytotoxicity.

### 3.3. Expression of SLC7A11 Is Up-Regulated by Selenite, Accompanied by Cystine Accumulation, Cysteine Reduction and NADPH Depletion in the Context of Glucose Deprivation

As mentioned above, SLC7A11-mediated uptake of extracellular cystine plays important role in regulating cellular redox homeostasis. We next asked whether the elevated ROS generation by the combination of selenite and glucose deprivation was associated with disruption of SLC7A11-regulated redox balance. We first examined the effect of selenite on SLC7A11 expression and results showed that exposure to selenite led to the up-regulation of SLC7A11 at all three levels of glucose, but stronger increase of SLC7A11 by selenite was found at concentration of 2.5 mM glucose (Figure 3A). In agreement with the increased SLC7A11 expression, intracellular level of cystine was significantly elevated at all three concentrations of glucose in response to selenite exposure (Figure 3B). Accordingly, intracellular level of cysteine was increased at concentrations of 25 mM and 10 mM glucose. In contrast, a dramatic reduction of intracellular level of cysteine was induced by selenite in the context of glucose deprivation (Figure 3C), indicating the conversion of cystine to cysteine was inhibited under such condition. The conversion of cystine to cysteine is a NADPH-dependent reaction, we therefore questioned whether shortage of NADPH contributed to the accumulation of cystine and reduction of cysteine. As shown in Figure 3D, glucose deprivation alone did not cause NADPH depletion, but a significant increased NADP+/NADPH ratio was detected with selenite exposure in the context of glucose deprivation, suggesting involvement of NADPH depletion in this redox imbalance. Consistent with the reduction of cysteine, intracellular level of GSH, a cysteine-based antioxidant, was significantly reduced by selenite under condition of glucose deprivation (Figure 3E). Together, these results indicated a good correlation between elevated SLC7A11 expression and redox collapse in response to selenite exposure in the context of glucose deprivation. In addition, the elevated expression of SLC7A11 was also observed in LLC and A549 lung cancer cells, HepG2 liver cancer cells and MDA-MB-231 breast cancer cells (Figure 3F). However, the up-regulation of SLC7A11 by selenite was not found in HK-2 normal kidney cells, suggesting that up-regulation of SLC7A11 by selenite is a specific event for cancer cells, which was consistent with the selective enhancement of selenite-mediated cytotoxicity of cancer cells. We also analyzed effect of MSeA on SLC7A11 expression and results showed that MSeA failed to induce up-regulation of SLC7A11, which was in line with lack of sensitization effect of glucose deprivation on MSeA.

### 3.4. SLC7A11-Mediated Cystine Accumulation and NADPH Depletion Contribute to the Elevated ROS Generation and Enhanced Cytotoxicity Induction by Combination of Selenite and Glucose Deprivation

To critically determine the functional role of selenite-mediated up-regulation of SLC7A11, we evaluated effect of SLC7A11 inhibition by its chemical inhibitor or knocking-down of SLC7A11 on selenite/glucose deprivation-induced cytotoxicity in HCT116 colon cancer cells. SAS, a known SLC7A11 inhibitor, was used to inactivate its function. Under such condition, the changes of cell viability were measured by crystal violet staining. As shown in Figure 4A–E, SLC7A11 inhibition by its inhibitor led to nearly abolishment of the cell viability reduction by the combination. Accordingly, ROS generation and cystine accumulation were blocked and the dysregulated NADP+/NADPH ratio and reduced GSH were recovered when SLC7A11 was inhibited by SAS. Similar results were also observed with another SLC7A11 inhibitor MSG (Figure 4F–H). Furthermore, the functional role of SLC7A11 was validated by siRNA approach and results showed that knocking-down of SLC7A11 mitigated the reduction of cell viability induced by selenite/glucose deprivation (Figure 4I). In addition, SAS and MSG also protected LLC cells from selenite/glucose deprivation combination-induced cytotoxicity (Appendix A). 2-Deoxy-glucose (2-DG), is commonly used as a glycolysis inhibitor. It can be phosphorylated by hexokinase (HK) to form 2-Deoxy-D-glucose-6-phosphate (2-DG-6-P) which cannot be shunted into the glycolysis pathway downstream of phosphoglucose isomerase, but can still be shunted into the PPP to produce NADPH [9,27]. We found that supplementation of 2-DG offered a significant protection on selenite/glucose deprivation-induced cytotoxicity (Figure 4J and Appendix A). Taken together, these results indicated that up-regulation of SLC7A11 by selenite played a pivotal role in the sensitization effect of glucose deprivation on selenite-induced cytotoxicity of cancer cells, which is attributed to SLC7A11-mediated cystine accumulation, NADPH depletion, GSH reduction and ROS generation.

### 3.5. Selenite/Glucose Deprivation-Induced Cytotoxicity Is Independent of Ferroptosis

Cysteine deprivation is one of common means to induce cell ferroptosis [28,29,30]. The above data showed that the cysteine level was dramatically reduced by selenite/glucose deprivation. We assumed that ferroptosis was involved in cytotoxicity induction by selenite/glucose deprivation. It has been shown that glutamine metabolism, known as glutaminolysis (conversion of glutamine to alpha-ketoglutarate), is required for cysteine deprivation-induced ferroptosis [31,32]. We first tested the effects of glutaminolysis inhibition by glutamine deprivation from medium on the cytotoxicity induction by selenite/glucose deprivation. As expected, glutamine deprivation reversed the cytotoxicity of selenite/low-glucose combination and reduced ROS level (Figure 5A,B and Appendix A). However, DFO or Fer-1, two ferroptosis inhibitors, failed to offer such protection (Figure 5C,D and Appendix A). Furthermore, no significant increase of LPO or LIP level (two biomarkers for ferroptosis) was detected in response to selenite/glucose deprivation (Figure 5E,F). These results suggested that ferroptosis was not involved in selenite/glucose deprivation-induced cytotoxic effect.

The above data demonstrated that glutamine was necessary for the enhanced cytotoxicity induced by selenite/glucose deprivation. Intracellular glutamate generated from glutamine metabolism and glutamine deprivation is supposed to cause reduction of glutamate, which might reduce SLC7A11-mediated uptake of cystine. We therefore speculated that the protection on the cytotoxicity offered by glutamine deprivation was attributed to its ability to inhibit cystine-glutamate exchange, which was similar to the condition of SLC7A11 inhibition by its inhibitor. To test this hypothesis, we measured the changes of intracellular cystine, GSH levels and NADP+/NADPH ratio in response to glutamine deprivation. Results showed that accumulated intracellular cystine and increased NADP+/NADPH ratio by combination of selenite and glucose deprivation were abolished, while GSH level was partially recovered under glutamine deprivation (Figure 5G–I), which was consistent with the reduced ROS (Figure 5B). These results provided a mechanistic explanation for the protection offered by glutamine deprivation. For most cancer cells, glucose deprivation may induce a dependence on glutamine for cell survival, making the cell more susceptible to glutamine deprivation. Our results indicated that glutamine deprivation delayed NADPH depletion and GSH reduction, thereby preventing redox collapse induced by selenite/low-glucose combination within the timeframe. However, for longer time of challenge, glutamine deprivation was expected to accelerate the cytotoxicity due to energy depletion. Therefore, timing is a possible reason for the paradoxical role of glutamine deprivation on cell survival.

### 3.6. Fasting Improves Therapeutic Efficacy of Selenite In Vivo

Having found the sensitization effect of glucose deprivation on selenite-induced cytotoxicity of cancer cells, we next asked whether the enhanced effect could be achieved in vivo. Fasting for 24 h results in a decrease in both plasma and intra-tumor glucose levels of mice [1,33]. Next, HCT116 and LLC xenograft models were employed to evaluate the anticancer effect of selenite/fasting described in Materials and Methods. As shown in Figure 6A,B, intermittent fasting caused a progressive inhibition of tumor growth and reduction of tumor weight, whereas selenite alone did not. Selenite/fasting combination further significantly delayed the tumor growth and reduced the tumor weight. Consistent with the in vitro findings, selenite/fasting combination decreased GSH levels and increased NADP+/NADPH ratio, which reflected the redox imbalance in selenite/fasting combination-treated tumors (Figure 6C,D). Moreover, SLC7A11 expression was significantly up-regulated in response to treatment with selenite alone or in combination with fasting (Figure 6E). The combination did not cause decrease of bodyweight in comparison with fasting alone, indicating no increased toxicity by the combination (data not shown). These results suggested that the fasting was able to improve efficacy of selenite against colon and lung cancer in xenograft models, which was associated with up-regulation of SLC7A11 expression and induction of oxidative stress.

## 4. Discussion

Sodium selenite has not really been considered as an appropriate choice of selenium compound for chemoprevention due to genotoxic potential from a range of toxicological studies [34,35]. The dose-limiting toxicity of selenite is still a major concern for its clinical use even though the selenophilic properties of cancer cells. Approaches that can potentiate cancer cells to selenite are clearly needed for promoting selenite as a clinical useful anticancer agent. In the present study, we demonstrated that the oxidative stress and cytotoxic effect induced by selenite were amplified in the context of glucose limitation, mechanistically associated with SLC7A11-mediated cystine accumulation, NADPH depletion, the conversion of cystine to cysteine inhibition and redox collapse. Moreover, this sensitization effect on cancer cells was not observed on the normal cells. Accordingly, the in vivo efficacy was significantly improved by the combination of selenite and fasting treatment without increased toxicity. The findings of the present study provide an effective and practical approach for increasing the therapeutic window of selenite and promoting the development of selenite as a clinically useful selective anticancer agent.

SLC7A11-mediated uptake of cystine plays an important role in maintaining redox homeostasis [36]. Conversion of cystine to cysteine is a critical step for cystine-mediated antioxidant function [37]. NADPH, a product of pentose phosphate pathway, is required for the conversion of cystine to cysteine to synthesize intracellular antioxidant glutathione (GSH) [38,39]. Elevated SLC7A11 has been implicated in malignant transformation and drug resistance [40,41,42,43]. In the present study, we demonstrated that this upregulated transporter rendered cancer cells susceptible to selenite in the context of glucose deprivation, indicating a double-edge-sword property of SLC7A11 for tumors. We found that SLC7A11 expression was up-regulated by selenite in all cancer cell lines tested either in glucose replete or condition of starvation. Up-regulation of SLC7A11 is supposed to promote cystine uptake, which is rapidly converted to cysteine, leading to elevated GSH levels in the context of glucose replete. Indeed, our data showed that selenite-induced SLC7A11 resulted in increase of cysteine and GSH under glucose sufficient condition. This SLC7A11-mediated increase of antioxidant capacity might compromise the oxidative stress and cytotoxic effect induced by selenite in the context of sufficient glucose. In contrast, under the condition of glucose limitation, the up-regulation of SLC7A11 by selenite was accompanied by increased cystine accumulation, decreased cysteine and GSH, followed by elevated ROS generation and cell death induction. In other words, glucose deprivation disrupted SLC7A11-cystine-cysteine-GSH-mediated antioxidant system, which in turn boosted selenite-induced oxidative stress and cytotoxic effect against cancer. NADPH availability is supposed to be responsible for this paradoxical role of SLC7A11 in regulating redox balance in different condition of glucose. Due to glucose is a major source for generating NADPH, the depletion of glucose is expected to reduce the availability of NADPH, thereby blocking the conversion of cystine to cysteine, which in turn led to enhanced ROS generation and cell death induction in response to selenite. Thus, the protective effect of 2-DG on selenite/glucose deprivation-induced cytotoxicity is likely attributed to its ability to produce NADPH through PPP pathway. On the contrary, glucose deprivation or BAY-876 sensitized cancer cells to selenite-induced cytotoxicity, because they directly cut off the glucose supply or uptake, which meant that there was not enough glucose available for the PPP to produce NADPH under the condition of glucose deprivation or BAY-876 treatment. It is worth pointing out that the sensitization effect of glucose limitation on cancer cells in response to selenite was not observed in normal cells, which was consistent with that the up-regulation of SLC7A11 by selenite in cancer cells but not found in normal cells, further supporting the role of SLC7A11 in glucose starvation-mediated sensitization effect on cancer cells. The mechanisms underlying the differential effect of selenite on SLC7A11 expression between cancer cells and normal cells needs further investigation.

Oxidative stress induction by selenite is well established, which is suggested to play a pivotal role in its cytotoxic effect on cancer cells. It has been shown that oxidative stress induction can sensitize cancer cells to a variety of anticancer agents and radiotherapy [8,44,45]. Our previous study has shown that selenite sensitized LNCaP prostate cancer cells to TRAIL-induced apoptosis, which is due to its ability to generate ROS [46]. A sensitization effect of selenite on refractory prostate cancer cells to radiation is also associated with the redox status [19]. The induction of oxidative stress by selenite was boosted in the context of glucose starvation, the findings therefore implied that utilizing IF and selenite in combination with radiation or chemotherapy would be expected to improve the therapeutic efficacy in the clinical setting.

As mentioned above, the form of selenium is an important factor affecting its anticancer activity. In the preventive setting, Larry Clark et al. [22] demonstrated that supplementation of selenium (Se) in the form of selenized yeast, which contains multiple forms of selenium compounds, leads to reduction of cancer risk, especially the cancer of prostate, lung and colon. However, a large-scale human intervention with selenomethionine (SeMet) supplementation (the selenium and vitamin E cancer prevention trial, SELECT) in North America failed to achieve an inhibitory effect on prostate carcinogenesis [47]. One possible reason for these controversial outcomes is an incorrect choice of selenium form for this clinical study in a context of selenium-adequate condition. This notion is supported by our previous study, in which, MSeA, but not selenite or SeMet is effective in a xenograft model of prostate cancer [48]. In the present experimental setting, our data showed that selenite alone was still ineffective, but combining glucose limitation with selenite produced a significantly enhanced anticancer effect in the two xenograft models. The sensitization effect was not induced by combining glucose limitation with MSeA. These data support a context-dependent nature of selenium compound-mediated anticancer effect. The sensitization effect of glucose limitation on selenite requires SLC7A11-mediated redox collapse. This event was not observed in response to MSeA, providing possible interpretation for the lack of sensitization effect of glucose limitation on MSeA. These results are consistent with the findings from previous reports that MSeA is metabolized to methylselenol to trigger caspase-dependent apoptosis without induction of superoxide formation and DNA damage, whereas selenite induces caspase-independent apoptosis and autophagic cell death associated with superoxide generation [49].

## 5. Conclusions

In summary, SLC7A11 was up-regulated by selenite exposure both in vitro and in vivo. The elevated expression of SLC7A11 by selenite resulted in cystine accumulation, NADPH depletion and redox collapse in the context of glucose starvation. Under such condition, the cytotoxic effect of selenite on cancer cells was specifically enhanced. Moreover, the therapeutic efficacy of selenite in vivo was greatly improved in xenograft models of lung and colon cancer when the treatment was coupled with fasting. The findings of the present study suggest that combination selenite and fasting holds promising potential to be developed as an effective treatment regimen for certain types of cancer.

## Figures and Tables

**Figure 1 cancers-14-00345-f001:**
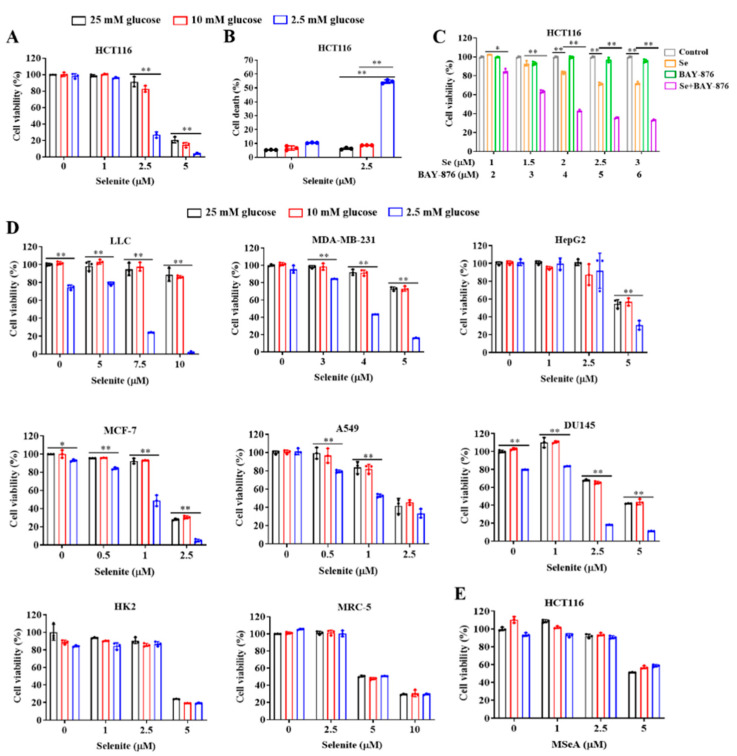
Glucose limitation sensitizes cancer cells to selenite-mediated cytotoxic effect. (**A**,**B**) Cell viability (**A**) and cell death (**B**) of HCT116 cells cultured in the medium containing indicated concentrations of glucose with or without treatment of selenite for 24 h. (**C**) Cell viability of HCT116 cells with or without treatment of selenite, BAY-876 or selenite/BAY-876 combination for 48 h. (**D**) Cell viability of LLC, A549, HepG2, MCF-7, MDA-MB-231, DU145, HK2 and MRC-5 cells cultured in the medium containing indicated concentrations of glucose with or without treatment of selenite for 24 h, 36 h, 36 h, 48h, 36 h, 48 h, 48 h and 48 h, respectively. (**E**) Cell viability of HCT116 cells cultured in the medium containing indicated concentrations of glucose with or without treatment of methylseleninic acid for 24 h. Results are representative of three biologically independent experiments. Data are expressed as mean ± SD, * *p* < 0.05, ** *p* < 0.01. Se: selenite; MSeA: methylseleninic acid.

**Figure 2 cancers-14-00345-f002:**
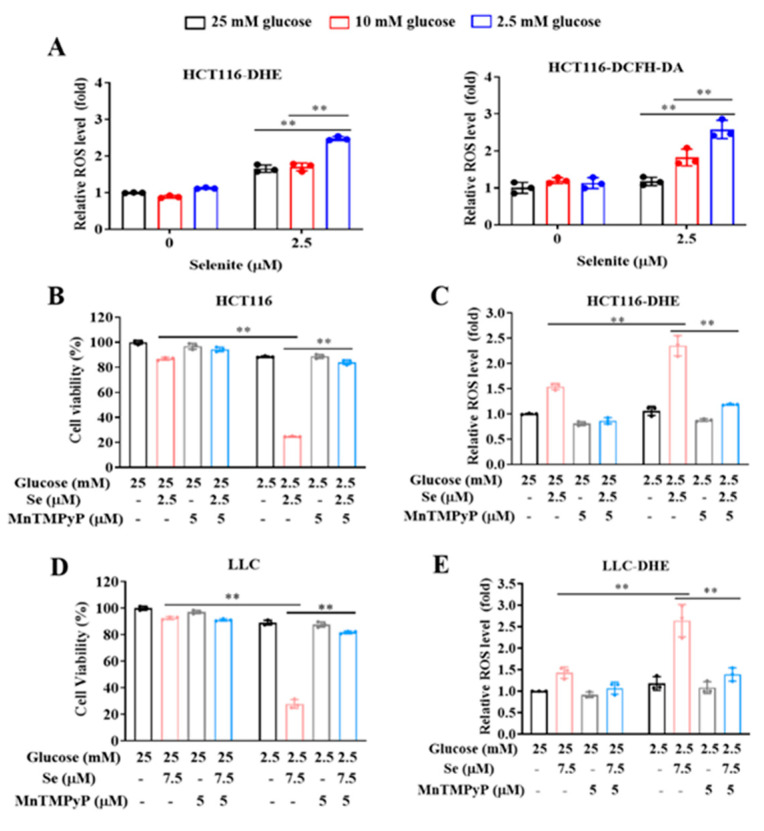
The sensitization effect of glucose limitation on selenite is attributed to elevated ROS generation. (**A**) The intracellular ROS level staining with DHE (for O_2_^•−^) or DCFH-DA (for H_2_O_2_) of HCT116 cells cultured in the medium containing indicated concentrations of glucose with or without treatment of selenite for 22 h. (**B**) Cell viability of HCT116 cells cultured in the medium containing 25 mM or 2.5 mM glucose with or without treatment of selenite or MnTMPyP for 24 h. (**C**) O_2_^•−^ levels of HCT116 cells cultured in the medium containing 25 mM or 2.5 mM glucose with or without treatment of selenite or MnTMPyP for 22 h. (**D**) Cell viability of LLC cells cultured in the medium containing 25 mM or 2.5 mM glucose with or without treatment of selenite or MnTMPyP for 24 h. (**E**) O_2_^•−^ levels of LLC cells cultured in the medium containing 25 mM or 2.5 mM glucose with or without treatment of selenite or MnTMPyP for 22 h. Results are representative of three biologically independent experiments. Data are expressed as mean ± SD, ** *p* < 0.01. MnTMPyP: manganese (III) tetrakis (1-methyl-4-pyridyl) porphyrin.

**Figure 3 cancers-14-00345-f003:**
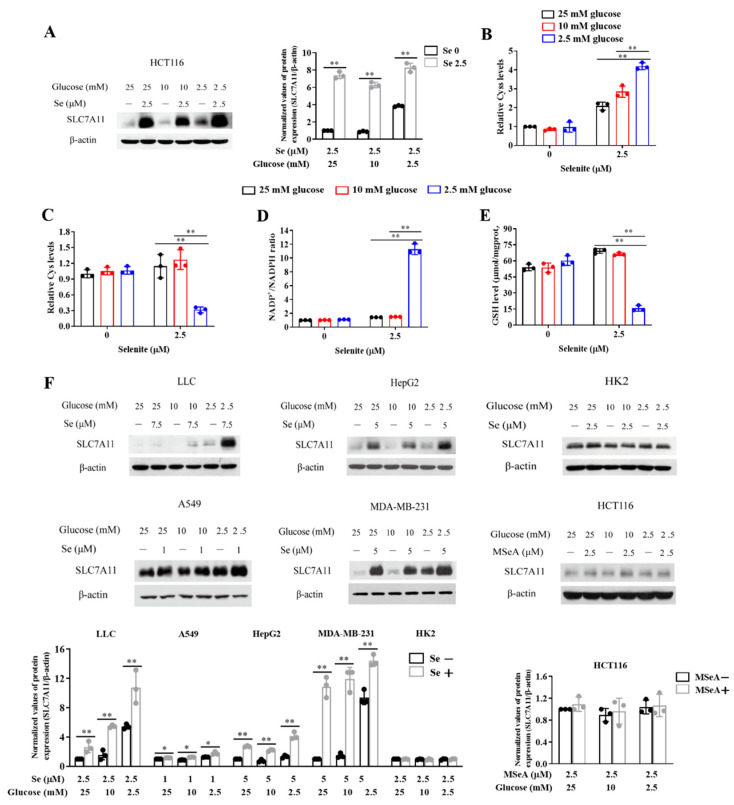
Expression of SLC7A11 is up-regulated by selenite, accompanied by cystine accumulation, cysteine reduction and NADPH depletion in the context of glucose deprivation. (**A**) SLC7A11 protein levels of HCT116 treated with or without selenite in different concentrations of medium glucose for 22 h. (**B**–**E**) Cyss levels (**B**), Cys levels (**C**), NADP+/NADPH ratios (**D**) and GSH levels (**E**) of HCT116 cells cultured in the medium containing indicated concentrations of glucose with or without treatment of selenite for 22 h. (**F**) SLC7A11 protein levels of LLC, A549, MDA-MB-231, HepG2 and HK2 cells treated with or without selenite in different concentnations of medium glucose for 22 h, 30 h, 30 h, 30 h and 48 h, respectively; SLC7A11 protein levels of HCT116 cells treated with or without MSeA in different concentnations of medium glucose for 22 h. Results are representative of three biologically independent experiments. Data are expressed as mean ± SD, * *p* < 0.05, ** *p* < 0.01. Cyss: cystine; Cys: cysteine; NADP+: oxidized form of nicotinamide-adenine dinucleotide phosphate; NADPH: reduced form of nicotinamide-adenine dinucleotide phosphate; GSH: glutathione; Se: selenite; MSeA: methylseleninic acid.

**Figure 4 cancers-14-00345-f004:**
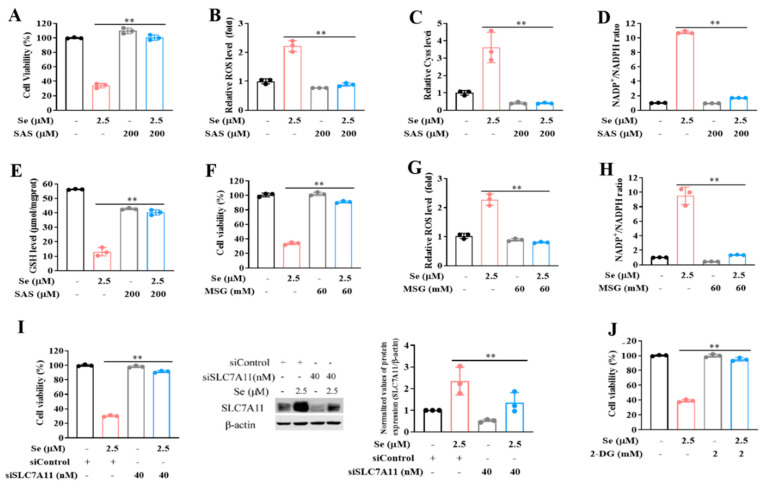
SLC7A11-mediated cystine accumulation and NADPH depletion contribute to the elevated ROS generation and enhanced cytotoxicity induction by combination of selenite and glucose deprivation. (**A**–**E**) Cell viability (**A**), O_2_^•−^ levels (**B**), Cyss levels (**C**), NADP+/NADPH ratios (**D**) and GSH levels (**E**) of HCT116 cells cultured in the medium containing 2.5 mM glucose with or without treatment of selenite or SAS for 24 h (**A**) and 22 h (**B**–**E**). (**F**–**H**) Cell viability (**F**), O_2_^•−^ levels (**G**) and NADP+/NADPH ratios (**H**) of HCT116 cells cultured in the medium containing 2.5 mM glucose with or without treatment of selenite or MSG for 24 h (**F**) and 22 h (**G**,**H**). (**I**) Cell viability and SLC7A11 protein levels of HCT116 cells expressing either scrambled siRNA or siRNA against SLC7A11 cultured in the medium containing 2.5 mM glucose with or without treatment of selenite for 24 h and 22 h, respectively. (**J**) Cell viability of HCT116 cells cultured in the medium containing 2.5 mM glucose with or without treatment of selenite or 2-DG for 24 h. Results are representative of three biologically independent experiments. Data are expressed as mean ± SD, ** *p* < 0.01. Se: selenite; SAS: salicylazosulfapyridine; Cyss: cystine; NADP+: oxidized form of nicotinamide-adenine dinucleotide phosphate; NADPH: reduced form of nicotinamide-adenine dinucleotide phosphate; GSH: glutathione; MSG: monosodium glutamate; 2-DG: 2-deoxy-D-glucose.

**Figure 5 cancers-14-00345-f005:**
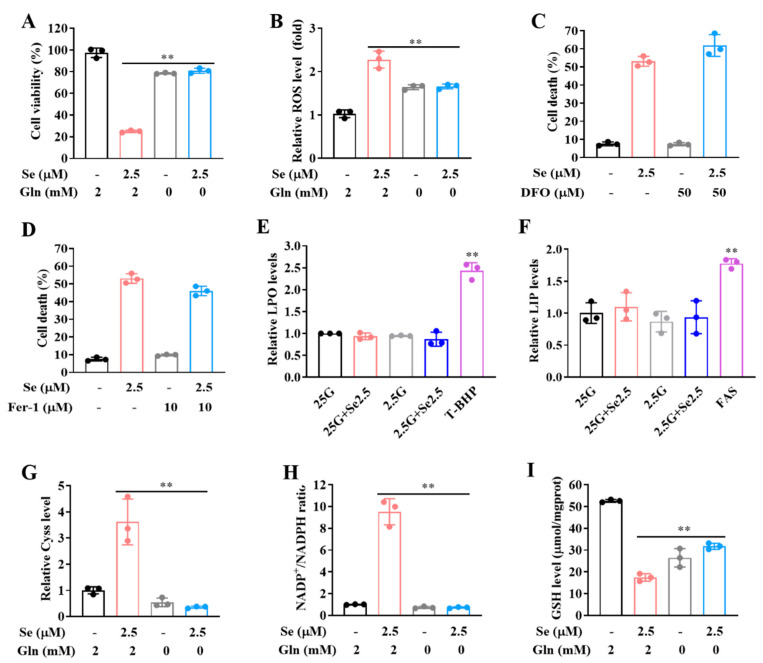
Selenite/glucose deprivation-induced cytotoxicity is independent of ferroptosis. (**A**,**B**) Cell viability (**A**) and O_2_^•−^ levels (**B**) of HCT116 cells cultured in the medium containing 2.5 mM glucose with or without treatment of selenite or glutamine deprivation for 24 h (**A**) and 22 h (**B**). (**C**,**D**) Cell death of HCT116 cells cultured in the medium containing 2.5 mM glucose with or without treatment of selenite or DFO or Fer-1 for 24 h. (**E**,**F**) LPO levels (**E**) and LIP levels (**F**) of HCT116 cells cultured in the medium containing 2.5 mM glucose with or without treatment of selenite for 22 h. Cells cultured in the medium containing 2.5 mM glucose were treated by 1 mM T-BHP or 200 μM FAS for 30 min before detection. (**G**–**I**) Cyss levels (**G**), NADP+/NADPH ratios (**H**) and GSH level (**I**) of HCT116 cells cultured in the medium containing 2.5 mM glucose with or without treatment of selenite or glutamine deprivation for 22 h. Results are representative of three biologically independent experiments. Data are expressed as mean ± SD, ** *p* < 0.01. Gln: glutamine; DFO: deferoxamine mesylate; Fer-1: ferrostatin-1; LPO: lipid peroxidation; LIP: labile iron pool; T-BHP: tert-butyl hydroperoxide; FAS: ferrous ammonium sulfate; Cyss: cystine; NADP+: oxidized form of nicotinamide-adenine dinucleotide phosphate; NADPH: reduced form of nicotinamide-adenine dinucleotide phosphate; GSH: glutathione.

**Figure 6 cancers-14-00345-f006:**
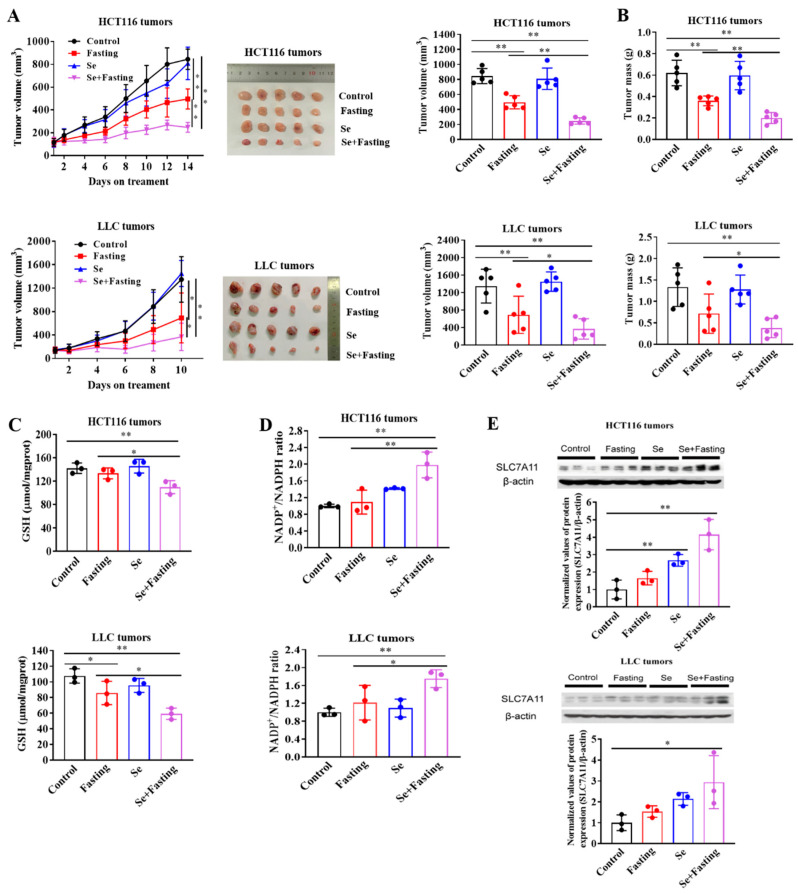
Fasting improves therapeutic efficacy of selenite in vivo. (**A**) In vivo growth of tumors as measured by tumor volume in mice inoculated with HCT116 cells or LLC cells; picture and volume of tumors isolated from mice in different groups. (**B**) Weight of tumors isolated from mice in different groups. (**C**–**E**) Measurement of GSH levels (**C**), NADP+/NADPH ratios (**D**) and SLC7A11 protein levels (**E**) of tumors isolated from mice in different groups. Data are expressed as mean ± SD, * *p* < 0.05, ** *p* < 0.01. Control: control group, mice were fed ad libitum and treated with water; Fasting: fasting group, mice experienced feeding/fasting cycles for 24 h, respectively; Se: selenite group, mice were fed ad libitum and treated with 2 mg/kg body weight of selenite during fasting cycle; Se + Fasting: selenite-hypoglycemia combination group, mice experienced feeding/fasting cycles for 24 h, respectively and were treated with 2 mg/kg body weight of selenite during fasting cycle, n = 5 per group.

## Data Availability

Data is contained within the article or Appendix A.

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
