# Peer review of "Glucose Limitation Sensitizes Cancer Cells to Selenite-Induced Cytotoxicity via SLC7A11-Mediated Redox Collapse"

_cancers, 2022, doi:10.3390/cancers14020345_

Round 1

Reviewer 1 Report

The study aimed to evaluate the cytotoxicity of selenite in cancer cells under glucose limitation in vitro and in vivo. However, the authors did not clearly define the selenite-induced cytotoxic effect as cell growth inhibition or cell death. Based on the method of crystal violet staining, the level of crystal violet will be affected by cell growth arrest and cell death, the two cytotoxic effects. The methodologies used in this study seem not to be suitable for the experimental designs and several data are questionable. The present data do not provide sufficient evidence to support the authors’ conclusion. Several concerns and defects are detected in the present study and should be addressed.

Major concerns:

  1. The questionable data in Figure 1 are that low glucose (2.5 mM) seems not to reduce cell growth in most cancer cells, such as HCT116, MDA-MB-231, HepG2, MCF7, and A549. Based on the method of crystal violet staining, the level of crystal violet will be affected by cell growth arrest and cell death, the two cytotoxic effects. As the mention for Warburg effect by the authors, these cancer cells should be relied on glucose for cell growth. However, the Figure 1 data seems not to indicate this effect. The authors should explain this issue.
  2. The images of b-actin for MDA-MB-231 and HCT116 in Figure 3F are not similar to those in Supplementary file.
  3. The questionable data in Figure 4J are that as a competitive inhibitor of glycolysis, the effect of 2-deoxy-glucose should be similar to those of the glucose deprivation or BAY-876. The data shown in Figure 4K that 2-deoxy-glucose exhibit different effects from the effects of glucose deprivation or BAY-876. The authors should address this issue.
  4. The questionable data in Figure 5AB are that glutamine deprivation reversed the cytotoxicity of selenite/low-glucose combination. For most cancer cells, glucose deprivation may induce a dependence on glutamine for cell survival, but the authors found that the glutamine deprivation reversed the cytotoxicity of selenite/low-glucose combination. The authors should explain this issue.
  5. The questionable data in Figure 5CD are that it is not clear whether selenite/glucose deprivation-induced cytotoxicity is independent of ferroptosis. For the experiments on ferroptosis, the authors used crystal violet staining rather than Annexin V/PI staining. The level of crystal violet will be affected by cell growth and cell death. Cell growth arrest and cell death will result in the relative low level of crystal violet staining. It is suggested that these experiments should be analyzed using PI exclusion assay for cell death determination under DFO and Fer-1 treatments.
  6. Figure 5E and 5F, the experimental conditions mentioned in figure legend seem not consistent with the figures. The detail experimental conditions for the two figures should be clearly mentioned.
  7. For the mechanism, the data in Figure 2, Figure 4, and Figure 5 are obtained from only one cell line. It is not clear whether the findings are cell specific manners or not. It is suggested that at least 3 different cancer cell lines should be examined in Figure 2, Figure 4, and Figure 5.
  8. For the in vivo study, it is not clear what glucose and glutamine levels are in the blood and tumors of these animals after treatments. Did the fasting treatment reduce glucose levels of blood and tumor? Did the fasting treatment repress tumor growth via reduced glucose level of blood and tumor? Based on the authors’ in vitro findings, only very low level of glucose could enhance the selenite-mediated cytotoxicity. Low glucose even did not affect cell growth of these cancer cells (See Figure 1, and Concern 1). However, in the in vivo experiments, fasting alone exerts significantly repressed tumor growth. The results should not be simply explained by glucose limitation effect. It is suggested that the authors should provide these data and give reasonable interpretations.

Minor concerns:

  1. Line 176: 2x105; Line 205: 2x106; Line 207: 3x106
  2. Line 304: 25Mm
  3. 14. [14]

Author Response

List of Responses

Major concerns:

Comment 1: The questionable data in Figure 1 are that low glucose (2.5 mM) seems not to reduce cell growth in most cancer cells, such as HCT116, MDA-MB-231, HepG2, MCF7, and A549. Based on the method of crystal violet staining, the level of crystal violet will be affected by cell growth arrest and cell death, the two cytotoxic effects. As the mention for Warburg effect by the authors, these cancer cells should be relied on glucose for cell growth. However, the Figure 1 data seems not to indicate this effect. The authors should explain this issue.

Response 1: The cell lines tested in the present study demonstrated different sensitivity to glucose limitation. MDA-MB-231, HepG2, MCF7, and A549 cells are relatively insensitive to glucose limitation in comparison with LLC cells. For these cell lines, a significant reduction of cell viability was detected when the cells were challenged with glucose limitation for an additional 12 h.

Comment 2: The images of b-actin for MDA-MB-231 and HCT116 in Figure 3F are not similar to those in Supplementary file.

Response 2: We apologize for the oversight! The mistakes have been corrected.

Comment 3: The questionable data in Figure 4J are that as a competitive inhibitor of glycolysis, the effect of 2-deoxy-glucose should be similar to those of the glucose deprivation or BAY-876. The data shown in Figure 4K that 2-deoxy-glucose exhibit different effects from the effects of glucose deprivation or BAY-876. The authors should address this issue.

Response 3: Thanks for the comment! As a glycolysis inhibitor, 2-deoxy-glucose blocks glycolysis and cannot be shunted into the glycolysis pathway downstream of phosphoglucose isomerase, but it can still be shunted into the PPP to produce NADPH (ref 1,2). Thus, 2-deoxy-glucose indirectly provides NADPH to protect cells from selenite/glucose deprivation-induced cytotoxicity. Glucose deprivation or BAY-876 directly cut off the glucose supply or uptake, which means that there is no enough glucose available for the PPP to produce NADPH under the condition of glucose deprivation or BAY-876 treatment.

References:

1. D Zhang, J Li, F Wang, J Hu, S Wang, Y Sun: 2-Deoxy-D-glucose targeting of glucose metabolism in cancer cells as a potential therapy. Cancer Lett 2014, 355:176-83.

2. X Liu, K Olszewski, Y Zhang, EW Lim, J Shi, X Zhang, J Zhang, H Lee, P Koppula, G Lei, et al: Cystine transporter regulation of pentose phosphate pathway dependency and disulfide stress exposes a targetable metabolic vulnerability in cancer. Nat Cell Biol 2020, 22:476-486.

Comment 4: The questionable data in Figure 5AB are that glutamine deprivation reversed the cytotoxicity of selenite/low-glucose combination. For most cancer cells, glucose deprivation may induce a dependence on glutamine for cell survival, but the authors found that the glutamine deprivation reversed the cytotoxicity of selenite/low-glucose combination. The authors should explain this issue.

Response 4: Thanks for the comment! Timing is a likely reason for the paradoxical role of glutamine deprivation. Our data demonstrated that deprivation for 24 h did not produce obvious cytotoxicity, conversely, it offered protection on selenite/low-glucose combination-induced cytotoxicity, which was corrected with elevated NADPH and GSH. These results indicated that glutamine deprivation delayed NADPH depletion and GSH reduction, thereby preventing redox collapse induced by selenite/low-glucose combination within the timeframe. However, for longer time of challenge, glutamine deprivation is expected to accelerate the cytotoxicity due to energy depletion. 

Comment 5: The questionable data in Figure 5CD are that it is not clear whether selenite/glucose deprivation-induced cytotoxicity is independent of ferroptosis. For the experiments on ferroptosis, the authors used crystal violet staining rather than Annexin V/PI staining. The level of crystal violet will be affected by cell growth and cell death. Cell growth arrest and cell death will result in the relative low level of crystal violet staining. It is suggested that these experiments should be analyzed using PI exclusion assay for cell death determination under DFO and Fer-1 treatments.

Response 5: Thanks for the suggestion! The requested data have been added.

Comment 6: Figure 5E and 5F, the experimental conditions mentioned in figure legend seem not consistent with the figures. The detail experimental conditions for the two figures should be clearly mentioned.

Response 6: Thanks for the suggestion! The experimental conditions of Figure 5E and 5F in figure legend have been corrected.

Comment 7: For the mechanism, the data in Figure 2, Figure 4, and Figure 5 are obtained from only one cell line. It is not clear whether the findings are cell specific manners or not. It is suggested that at least 3 different cancer cell lines should be examined in Figure 2, Figure 4, and Figure 5.

Response 7: Thanks for the suggestion! Additional cell line has been examined and the results are shown in Figure 2 and Figure S1.

Comment 8: For the in vivo study, it is not clear what glucose and glutamine levels are in the blood and tumors of these animals after treatments. Did the fasting treatment reduce glucose levels of blood and tumor? Did the fasting treatment repress tumor growth via reduced glucose level of blood and tumor? Based on the authors’ in vitro findings, only very low level of glucose could enhance the selenite-mediated cytotoxicity. Low glucose even did not affect cell growth of these cancer cells (See Figure 1, and Concern 1). However, in the in vivo experiments, fasting alone exerts significantly repressed tumor growth. The results should not be simply explained by glucose limitation effect. It is suggested that the authors should provide these data and give reasonable interpretations.

Response 8: Fasting is supposed to cause decrease in blood glucose level, which is supported by the literature (1.2). As we responded to question 1, it takes time to produce significantly inhibitory effect on cancer cell growth in vitro. Similar to the findings in vitro, the inhibitory effect of fasting on tumor growth was progressively induced. We will further investigate whether additional mechanisms are involved in the tumor growth inhibition in addition to glucose insufficiency in the future studies.

References:

1. M Elgendy, M Cirò, A Hosseini, J Weiszmann, L Mazzarella, E Ferrari, R Cazzoli, G Curigliano, A DeCensi, B Bonanni, et al: Combination of hypoglycemia and metformin impairs tumor metabolic plasticity and growth by modulating the PP2A-GSK3β-MCL-1 axis. Cancer Cell 2019, 35:798-815.

2. TL Jensen, MK Kiersgaard, LF Mikkelsen, DB Sørensen: Fasting of male mice – Effects of time point of initiation and duration on clinical chemistry parameters and animal welfare. Lab Anim-Uk 2019, 53:587-597.

Minor concerns:

  1. Line 176: 2x105; Line 205: 2x106; Line 207: 3x106
  2. Line 304: 25Mm
  3. 14. [14]

Response: Thanks for the reviewer’s careful reading! We have corrected in the revised manuscript.

Reviewer 2 Report

Brief summary

The impact of glucose levels on selenite induced cytotoxicity was evaluated in range of cancer and non-malignant cell lines. The role of SLC7A11 (cystine/glutamate antiporter) was highlighted as a potential target for modulation of selenite’s anticancer properties by manipulating glucose concentrations.  Further experiments showed the combination of low glucose and selenite to be effective at inducing cell death in range of cancer cell lines, while sparing non-malignant cells. This cell death was associated with cystine accumulation, NADPH and GSH depletion. In vivo assessment in mouse xenograft models confirmed the utility of intermittent fasting on selenite cytotoxicity in HCT116 and LLC tumours.

Broad comments.

This overall study hypothesis appears to be well-reasoned and the sequence of in vitro and in vivo assessments are logical in sequence and look to be conducted using appropriate methodologies. The overall manuscript could be improved with the judicious use of colour in the initial figures (especially figure 1). General improvements in sentence construction such as proper use of tenses would allow for greater accessibility. The role of glucose limitation and selenite in combination with established anti-cancer treatments would be really interesting.

With respect to the discussion, I would suggest that sodium selenite hasn’t really been considered as an appropriate choice of selenium compound for chemoprevention due to genotoxic potential from a range of toxicological studies.

Perhaps a greater focus on utilising IF and selenite in combination/ sequence with radiation/ chemo would be of more value in the clinical setting.

The review by Wallenberg looks at the differing forms of cell death triggered a range of Se compounds and might contribute to the discussion around seleno- compound specific activity in this setting.

Specific comments:

Title : ? replace cytotoxic effect with cytotoxicity.

Line 18 , 20,26, 29,34 grammar

Line 70, properties or characteristics/ rather than feature?

Line 71 cells

Line 315 good correlation rather than well

Line 440 and 441 re-phrase

457 condition of starvation

Line 468 re-phrase

Line 478 needs or requires further investigation.

Line 506 typo mediated.

Author Response

List of Responses

Comment 1: This overall study hypothesis appears to be well-reasoned and the sequence of in vitro and in vivo assessments are logical in sequence and look to be conducted using appropriate methodologies. The overall manuscript could be improved with the judicious use of colour in the initial figures (especially figure 1). General improvements in sentence construction such as proper use of tenses would allow for greater accessibility. The role of glucose limitation and selenite in combination with established anti-cancer treatments would be really interesting.

Response 1: Thanks for the reviewer’s positive comments and constructive suggestion! The figures have been revised according to the suggestion. In addition, we have carefully proofread the manuscript to minimize typos and grammar errors.

Comment 2: With respect to the discussion, I would suggest that sodium selenite hasn’t really been considered as an appropriate choice of selenium compound for chemoprevention due to genotoxic potential from a range of toxicological studies.

Perhaps a greater focus on utilizing IF and selenite in combination/ sequence with radiation/ chemo would be of more value in the clinical setting.

The review by Wallenberg looks at the differing forms of cell death triggered a range of Se compounds and might contribute to the discussion around seleno- compound specific activity in this setting.

Response 2: Thanks for the suggestion! The discussion section has been revised according to the reviewer’s suggestion.

Comment 3: Specific comments:

Title : ? replace cytotoxic effect with cytotoxicity.

Line 18 , 20,26, 29,34 grammar

Line 70, properties or characteristics/ rather than feature?

Line 71 cells

Line 315 good correlation rather than well

Line 440 and 441 re-phrase

Line 457 condition of starvation

Line 468 re-phrase

Line 478 needs or requires further investigation.

Line 506 typo mediated.

Response 3: Thanks for the reviewer’s suggestion and careful reading! These issues mentioned above have been corrected as the reviewer suggested.

Reviewer 3 Report

Overall impression 

The study by Chen et al has investigated for the first time the possibility of potentiating the ability of selenite to kill cancer cells. This is important as the toxicity of selenite represents a potentially limiting factor in its wider use as an anticancer therapeutic. 

The authors demonstrate that selenite dramatically induced expression of SLC7A11 which led to an increase in cystine entry and subsequent production of cysteine (a precursor of antioxidants, esp. GSH) under normal glucose conditions. Yet, and somewhat paradoxically, upon glucose deprivation the authors observed reduction in cysteine and GSH production, and elevation of ROS. It appears that this paradox is explained by the depletion of glucose, which reduces the availability of NADPH, blocks the conversion of the accumulating cystine to cysteine, thus sensitising cancer cells to selenite-mediated ROS production and cell death (which is not ferroptosis); notably, this sensitisation is not evident in non-malignant cells. Additional experimentation using mouse xenografts models (with two independent cell lines) provided good evidence for fasting being able to enhance the anti-tumour effects of selenite in vivo, thus corroborating the in vitro observations. Finally, the results strongly suggest that the effects observed are specific to selenite but not other forms of selenium.  

Overall, the study is well planned and appears to be soundly executed. There is certainly plenty of experimental data, and all conclusions made are supported by the results. And despite the graphs not utilising appropriate colour-scheme, the findings are compelling and may have important therapeutic value.   

Major points

1. An issue with all figures that contain bar graphs is that the colour scheme and style of the bars chosen make it difficult to clearly and readily identify the different conditions ([glucose]). I would thus strongly advise the authors to modify the styles used (e.g. use combination of empty, filled and hatched bars in the charts) for improved clarity. 

2. The manuscript suffers from grammatical issues and some awkward phrasing in places; thus, I would, respectfully, advise the authors to seek some assistance to improve language usage. I feel this would really enhance what is a good piece of work.  

Minor points

1. In theory the findings appear paradoxical, as SLC7A11 induction has been implicated in upregulation of antioxidant (GSH) pathways and thus facilitates malignant transformation. However, the lack of glucose and depletion of NADPH appears to cause accumulation of cystine and no cysteine production. Thus, it might be a useful addition to the manuscript to note (in the Discussion) how malignant transformation (e.g. Ras over-expression – Lim et al PNAS, 2019) appears to upregulate the SLC7A11 transporter to control oxidative stress. In light of the present study, this creates a ‘double-edge sword’ for tumours as the combination of selenite plus glucose deprivation now paradoxically renders these cells susceptible to the effects of this upregulated transporter.  

2. There are a couple of unfortunate issues with grammar and awkward phrasing in the ‘Simple Summary’ and ‘Abstract’. I do not mean to be pedantic or patronising in any way, but I felt it would be a shame not to draw the authors’ attention, as these are the summaries of the paper. More specifically:

a) In the Simple Summary and Line 7, “… form of selenium, and is preferentially…”: please remove the word “and”.

b) In the second sentence in the Abstract (lines 19-20), the final part of the sentence needs to be re-phrased, as it reads awkwardly. Perhaps say: “… form of selenium, that is preferentially accumulated…”?  

3. As mentioned above, the text would benefit from some grammatical improvement in many places as well as correction of typos. Examples of such issues include: 

a) Line 41: replace “its” with “their” 

b) Line 106: “noticed” should be “noted”

c) Line 140: please correct “cystetine” 

d) Line 315: “well” should be “good”  

e) Line 484: I assume “TRIAL” should be “TRAIL”?  

f) Line 506: correct to “SLC7A11-mediated”.  

Author Response

List of Responses

Major points

Comment 1: An issue with all figures that contain bar graphs is that the colour scheme and style of the bars chosen make it difficult to clearly and readily identify the different conditions ([glucose]). I would thus strongly advise the authors to modify the styles used (e.g. use combination of empty, filled and hatched bars in the charts) for improved clarity. 

Response 1: Thanks for the reviewer’s constructive suggestion! We have updated the figures based on the suggestion.

Comment 2: The manuscript suffers from grammatical issues and some awkward phrasing in places; thus, I would, respectfully, advise the authors to seek some assistance to improve language usage. I feel this would really enhance what is a good piece of work.  

 Response 2: Thanks for the reviewer’s suggestion! Done as suggested.

Minor points

Comment 3: In theory the findings appear paradoxical, as SLC7A11 induction has been implicated in upregulation of antioxidant (GSH) pathways and thus facilitates malignant transformation. However, the lack of glucose and depletion of NADPH appears to cause accumulation of cystine and no cysteine production. Thus, it might be a useful addition to the manuscript to note (in the Discussion) how malignant transformation (e.g. Ras over-expression – Lim et al PNAS, 2019) appears to upregulate the SLC7A11 transporter to control oxidative stress. In light of the present study, this creates a ‘double-edge sword’ for tumours as the combination of selenite plus glucose deprivation now paradoxically renders these cells susceptible to the effects of this upregulated transporter.  

Response 3: Thanks for the reviewer’s suggestion! We have updated the discussion as the reviewer suggested.

Comment 4: There are a couple of unfortunate issues with grammar and awkward phrasing in the ‘Simple Summary’ and ‘Abstract’. I do not mean to be pedantic or patronising in any way, but I felt it would be a shame not to draw the authors’ attention, as these are the summaries of the paper. More specifically:

  1. a) In the Simple Summary and Line 7, “… form of selenium, and is preferentially…”: please remove the word “and”.
  2. b) In the second sentence in the Abstract (lines 19-20), the final part of the sentence needs to be re-phrased, as it reads awkwardly. Perhaps say: “… form of selenium, that is preferentially accumulated…”?  
  3. As mentioned above, the text would benefit from some grammatical improvement in many places as well as correction of typos. Examples of such issues include: 
  4. a) Line 41: replace “its” with “their” 
  5. b) Line 106: “noticed” should be “noted”
  6. c) Line 140: please correct “cysteine” 
  7. d) Line 315: “well” should be “good”  
  8. e) Line 484: I assume “TRIAL” should be “TRAIL”?  
  9. f) Line 506: correct to “SLC7A11-mediated”.  

Response 4: Thanks for the reviewer’s suggestion and careful reading! These issues mentioned above have been corrected as the reviewer suggested.

Round 2

Reviewer 1 Report

Most concerns have been improved, but the authors' explanations were not mentioned in the revised manuscript. These concerns may be also the readers have. It is suggested that the authors’ responses to concerns 1, 3, 4 and 8 should be mentioned in Results or Discussion sections.

Author Response

List of Responses

Comment: Most concerns have been improved, but the authors' explanations were not mentioned in the revised manuscript. These concerns may be also the readers have. It is suggested that the authors’ responses to concerns 1, 3, 4 and 8 should be mentioned in Results or Discussion sections.

Response:Thanks for reviewer’s suggestion! Done as the reviewer suggested.